# EFFICIENT GRADIENT CLIPPING METHODS IN DP-SGD FOR CONVOLUTION MODELS

## ABSTRACT

Differentially private stochastic gradient descent (DP-SGD) is a well-known method for training machine learning models with a specified level of privacy. However, its basic implementation is generally bottlenecked by the computation of the gradient norm (gradient clipping) for each example in an input batch. While various techniques have been developed to mitigate this issue, there are only a handful of methods pertaining to convolution models, e.g., vision models. In this work, we present three methods for performing gradient clipping that improve upon previous state-of-art methods. Two of these methods use in-place operations to reduce memory overhead, while the third one leverages a relationship between Fourier transforms and convolution layers. To demonstrate the numerical efficiency of our methods, we also present several benchmark experiments that compare against other algorithms.

## 1 INTRODUCTION

Differentially-private stochastic gradient descent (DP-SGD) is a common tool used to train machine learning models to protect sensitive information contained within individual training records (Abadi et al., 2016). However, general implementations of DP-SGD are bottlenecked by their gradient clipping step, whose runtime and memory costs scale linearly with the batch size times the number of model parameters. Our goal in this work is to develop three improved variants of the gradient clipping step that are substantially more efficient when applied to models with convolution layers.

*DP-SGD details*. The DP-SGD algorithm (Chaudhuri et al., 2011; Bassily et al., 2014) relies on the Gaussian mechanism and composition of differential privacy (Dwork et al., 2006; 2014) across iterations to privately compute the average of per-example gradients in a batch. At each iteration it operates by (i) bounding the *sensitivity* of each record within a batch to control and quantify the impact of any single record on the final model weights, and (ii) adding Gaussian noise proportional to the inverse of the batch size times the bound in (i). In particular, sensitivity is controlled by bounding per-example gradient norms so that the privatized gradients lie in a compact set. This approach is crucial for reducing noise growth, which scales as $\mathcal{O}(\sqrt{d}/[\epsilon b])$ (Bassily et al., 2014), where $d$ is the number of model parameters, $\epsilon$ the privacy budget, and $b$ the number of records in a batch. Alternatively, one can clip the overall average gradient at each step, but this increases noise by a factor of the batch size to $\mathcal{O}(\sqrt{d}/\epsilon)$.

Naive per-example clipping requires computing the norm of all per-example gradients. Specifically, this methods requires storing at least a matrix of size $\Theta(bd)$ that contains per-example gradients. Given the importance of model utility within this privacy-preserving context, there have been several developments on improving this step (with a focus on models with fully-connected or embedding layers). For example, techniques like ghost-clipping (Goodfellow, 2015) have been leveraged to improve both the runtime and storage complexity in certain settings. However, similar savings for convolution layers remain elusive (Rochette et al., 2019; Lee & Kifer, 2021a).

*Contributions*. This work introduces three novel gradient clipping methods that outperform prior methods for convolution methods in certain regimes. More specifically,

- the first two methods use in-place calculations and obtain $O(1)$ per-example storage complexities;

- the first method directly computes the squared norm, while the second leverages the ghost-clipping trick for fully-connected layers;

- the third method uses a relationship between convolution operators and fast Fourier transforms (FFTs) to obtain a scheme that scales well in the high-dimensional setting.

It is worth mentioning that the analysis for the third method (in the context of gradient clipping methods) appears to be new. In particular, this analysis exploits properties of circulant matrices to derive an algorithm that runs efficiently in terms of the number of model parameters $d$, and the batch size $b$.

To verify the practical efficiency of our methods, we also provide several benchmark experiments that demonstrate the numerical efficiency of our proposed methods.

*Related work.* The vast literature on DP-SGD (Chaudhuri et al., 2011; Bassily et al., 2014; Abadi et al., 2016; Ponomareva et al., 2023; Bu et al., 2023a) highlights the challenge of bounding individual record sensitivity, a crucial aspect often addressed through clipping[1]. While alternative approaches exist, such as modifying model architectures to enable Lipschitz constant computation (Béthune et al., 2023), their broader applicability remains uncertain.

To the best of our knowledge, the state-of-the-art performance in the setting of convolution models is achieved by Bu et al. (2023b). Specifically, that work builds on the approach of Bu et al. (2022); Lee & Kifer (2021a) and combines it with a careful book-keeping scheme that avoids a second back-propagation step. The main observation of Bu et al. (2022) is that the straightforward implementation of DP-SGD can be faster or more memory-efficient than ghost-clipping in certain regimes. While Rochette et al. (2019); Lee & Kifer (2021a) rely on instantiating per-example gradients, Bu et al. (2022) take advantage of the underlying network structure and choose which of of two different approaches to run; this selection step drives the bulk of their speed-up. In a follow-up work (Bu et al., 2023b), the authors use the previous observation and the idea that the second back-propagation step can be avoided using caching techniques.

Fourier transforms have first been used to improve the efficiency of training convolution neural networks (CNNs) by Mathieu et al. (2013), who build upon related work by Ben-Yacoub et al. (1999) for small-scale fully-connected models. Additional improvements to the approach have been developed, for example, by Pratt et al. (2017); Vasilache et al. (2014); Abtahi et al. (2017); Rippel et al. (2015). However, the development of similar techniques for the purpose of gradient clipping (this work) appears to be new.

For convenience, we compare in Table 1.1 the asymptotic time runtime and storage complexities of the methods by Bu et al. (2022), Lee & Kifer (2021a), and our proposed methods.

Table 1.1: *Asymptotic time and space complexities of various gradient clipping methods for a single example. The scalars $n_{\text{in}}$, $n_{\text{out}}$, $d_k$, $d_{\text{in}}$, and $d_{\text{out}}$ denote the number of input channels, output channels, kernel size, input dimension, and output dimension, respectively. Direct methods materialize the unaltered gradients, ghost-clipping methods apply the trick from Goodfellow (2015), and FFT methods utilize a novel relationship between convolution layers and FFTs proposed in this work.*

| Method | Type | Runtime | Storage |
|---|---|---|---|
| Lee & Kifer (2021b) | direct | $n_{\text{in}}n_{\text{out}}d_{\text{out}}d_k$ | $n_{\text{out}}d_{\text{out}} + n_{\text{in}}d_{\text{out}}d_k$ |
| Bu et al. (2022) | ghost-clipping | $d_{\text{out}}^2(n_{\text{in}}d_k + n_{\text{out}})$ | $d_{\text{out}}^2 + n_{\text{out}}d_{\text{out}} + n_{\text{in}}d_{\text{in}}d_k$ |
| Algorithm 3.1 [ours] | direct | $n_{\text{in}}n_{\text{out}}d_{\text{out}}d_k$ | $O(1)$ |
| Algorithm 3.2 [ours] | ghost-clipping | $d_{\text{out}}^2(n_{\text{in}}d_k + n_{\text{out}})$ | $O(1)$ |
| Algorithm 3.3 [ours] | FFT | $n_{\text{in}}n_{\text{out}}d_{\text{in}}\log(d_{\text{in}})$ | $d_{\text{in}}$ |

---

[1]See Pichapati et al. (2019); Chen et al. (2020) for examples or Ponomareva et al. (2023) for a recent overview.

*Notation.* For a matrices $A$ and $B$ we let $\|A\|$ denote the Frobenius norm of $A$ and $\langle A, B \rangle$ denote the (Frobenius) inner product. Let $(\mathcal{W}, \langle \cdot, \cdot \rangle)$ and $(\mathcal{Y}, \langle \cdot, \cdot \rangle)$ denote two Hilbert spaces with common induced norm $\| \cdot \|$. We denote linear operators between them by italicized letters $\mathcal{A} \colon \mathcal{W} \to \mathcal{Y}$ and denote $\mathcal{A}^* \colon \mathcal{Y} \to \mathcal{W}$ to be the *adjoint* of $\mathcal{A}$. That is, $\mathcal{A}^*$ is the unique linear operator that satisfies

$$\langle y, \mathcal{A}w \rangle = \langle \mathcal{A}^* y, w \rangle \quad \forall w \in \mathcal{W}, \quad \forall y \in \mathcal{Y}. \tag{1}$$

Let $\psi \colon \mathcal{W} \to \mathcal{Y}$ be an arbitrary function. The *Fréchet derivative* of $\psi$ at $w_0 \in \mathcal{W}$ is given by the unique bounded linear operator $D\psi(w_0) \colon \mathcal{W} \to \mathcal{Y}$ satisfying

$$\lim_{\delta \to 0} \frac{\|\psi(w_0 + \delta) - \psi(w_0) - D\psi(w_0)\delta\|}{\|\delta\|} = 0.$$

We say $\psi$ is differentiable if its Fréchet derivative exists for all $w_0 \in \mathcal{W}$. Throughout this paper we will use two special properties of the Fréchet derivative: the chain rule and the existence of gradients. Let $(\mathcal{Z}, \langle \cdot, \cdot \rangle)$ be another Hilbert space and $\phi \colon \mathcal{Y} \to \mathcal{Z}$ be given. The *chain rule* provides us with a simple way to calculate the derivative of the function $\phi \circ \psi \colon \mathcal{W} \to \mathcal{Z}$, namely,

$$D(\phi \circ \psi)(w_0) = D\phi(\psi(w_0))D\psi(w_0).$$

The Fréchet derivative of $\psi$ at $w_0$ with respect to a subset of variables $u$ is denoted by $D_u\phi(w_0)$. Finally, $\nabla\psi(w_0) \in \mathcal{W}$ denotes the (unique) gradient of a function $\psi$ at $w_0$, which satisfies

$$D\psi(w_0)\delta = \langle \nabla\psi(w_0), \delta \rangle_{\mathcal{W}} \quad \forall \delta \in \mathcal{W}. \tag{2}$$

The existence of the gradient is guaranteed by the well-known Riesz-Fréchet Representation Theorem (Rudin et al., 1976). The gradient of $\psi$ at $w_0$ with respect to a set of variables $u$ is denoted by $\nabla_u\psi(w_0)$.

*Organization.* Section 2 presents some necessary background material on representing gradient norms in convolution models. Section 3 presents the proposed clipping methods and discusses their properties and algorithm complexities under different regimes. Finally, Section 4 gives several numerical experiments and benchmarks.

## 2 BACKGROUND

To simplify our presentation, we focus on a single convolution layer and a single example $x \in \mathbb{R}^{n_{\mathrm{in}} \times d_{\mathrm{in}}}$ from the batch of inputs. For the case of multiple convolution layers and multiple examples, it is straightforward to see that our complexity results scale linearly with the number of layers times the number of examples. Moreover, we present our results for one-dimensional inputs; in Section 3.3 we discuss generalizations of our approaches to higher-dimensional inputs.

Given a stride length $s \geq 1$, let $d_k \in \mathbb{N}$, $d_{\mathrm{in}} \in \mathbb{N}$, $d_{\mathrm{out}} = 1 + (d_{\mathrm{in}} - d_k)/s$ be the size[2] of the kernel, inputs, and outputs, respectively, let $n_{\mathrm{in}} \in \mathbb{N}$ and $n_{\mathrm{out}} \in \mathbb{N}$ be the number of input, output channels, respectively, and let $w \in \mathbb{R}^{n_{\mathrm{in}} \times n_{\mathrm{out}} \times d_k}$ be the kernel weights. Moreover, for fixed output channel $j$, let (i) $w^{i,j} \in \mathbb{R}^{d_k}$ be the kernel vector corresponding to the $i$-th input channel, (ii) $b^j \in \mathbb{R}^{n_{\mathrm{out}} \times d_{\mathrm{out}}}$ be the bias offset, (iii) $\alpha$ be a general activation function, and (iv) $U_x^i \in \mathbb{R}^{d_{\mathrm{out}} \times d_k}$ be a matrix whose $\ell$-th row consists of the entries in the $i$-th input channel of $x$ that are being multiplied with $w^{i,j}$.

The output for the $j$-th output channel of a convolution layer is given by

$$[\phi_x(w,b)]^j = \phi_x^j(w,b) := \alpha\left(b^j + \sum_{i \in [n_{\mathrm{in}}]} U_x^i w^{i,j}\right). \tag{3}$$

Numerically efficient schemes for computing $\|\nabla_b \phi_x^j(w,b)\|^2$ (the bias weights' gradient norm), have been previously developed by Kong & Munoz Medina (2024). Consequently, our focus is on analyzing the kernel weights' gradient norm $\|\nabla_w \phi_x^j(w,b)\|^2$. Following similar analyses as Kong & Munoz Medina (2024), we first write

$$\phi_x^j = \ell_x \circ \psi_x^j \circ Z_x \quad \text{where} \quad \psi_x^j(z) := \alpha(z + b^j), \quad Z_x^j(w) := \sum_{i \in [n_{\mathrm{in}}]} U_x^i w^{i,j}. \tag{4}$$

---

[2]To avoid clutter, we assume these are all integers. In the implementation of our approach, we handle the general case.

Then, if we denote

$$\mathcal{A} = \mathcal{A}_x(w) := DZ_x^j(w), \qquad g^j = g_x^j(w) := \nabla(\ell_x \circ \psi_x^j)(Z_x^j(w)), \tag{5}$$

it follows from the chain rule that

$$\|\nabla_w \phi_x^j(w,b)\|^2 = \Omega_x(g^j) := \left\|\mathcal{A}^* g^j\right\|^2. \tag{6}$$

To avoid the notational clutter, we denote the adjoint operator of $DZ_x^j(w)$ by $DZ_x^{j*}(w)$. Using the fact that $\nabla_w \phi_x(w,b) = [\nabla_w \phi_x^1(w,b), \ldots, \nabla_w \phi_x^{n_{\text{out}}}(w,b)]$, we have that

$$\|\nabla_w \phi_x(w,b)\|^2 = \sum_{j=1}^{n_{\text{out}}} \|\nabla_w \phi_x^j(w,b)\|^2 = \sum_{j=1}^{n_{\text{out}}} \left\|\mathcal{A}^* g^j\right\|^2,$$

and, hence, it suffices to restrict our presentation to a fixed output channel $j$ where applicable. Kong & Munoz Medina (2024) established efficient representations of $\Omega_x(g)$ for the case of embedding and fully-connected layers. Similarly, our task will be to find an efficient representation of $\Omega_x(g)$ for convolution layers.

## 3 ALGORITHMS AND DISCUSSION

This section contains three subsection that present the main algorithms and technical discussion of our work. The first subsection presents the in-place algorithms and their properties, the second one presents the Fourier-based algorithm and its properties, and the last section compares various methods across different regimes, considering the number of input-output channels and input-output dimensions.

Before proceeding, we describe some common notation and a basic result about the function $Z_x(\cdot)$ in (5). Given a 4D array $M \in \mathbb{R}^{n_{\text{in}} \times n_{\text{out}} \times d_k \times d_{\text{out}}}$, we denote $M_{m,\ell}^{i,j}$ to be the value in the corresponding to the $i$-th input channel, $j$-th output channel, $m$-th input dimension, and $\ell$-th output dimension of $M$. We give similar definitions for the arrays/scalars $M^{i,j}$, $M_m^i$, $M_\ell^j$, $M^i$, and $M^j$, keeping the convention that superscripts (resp. subscripts) contain indices for the input/output channels (resp. dimensions). The straightforward representation of the operators we have discussed so far requires defining and handling fourth-order tensors, which can vastly complicate the analysis. However, we are able to decompose various operations across different channels and dimensions, which allows us to only use two-dimensional matrices to represent all the operators we use.

The result below provides some convenient representations of the Fréchet derivative of $Z_x^j(w)$ and $Z_x^{j*}(w)$. Its proof is postponed to Appendix A.

**Lemma 3.1.** *Let $U_x^i \in \mathbb{R}^{d_{\text{out}} \times d_k}$ be as in (4) for some input channel $i \in [n_{\text{in}}]$, let $\Delta \in \mathbb{R}^{n_{\text{in}} \times n_{\text{out}} \times d_k}$, and $\tau^j \in \mathbb{R}^{d_{\text{out}}}$ be arbitrary. If $\Delta^{i,j} \in \mathbb{R}^{d_k}$ is the displacement vector corresponding to input-output channel pair $(i,j) \in [n_{\text{in}}] \times [n_{\text{out}}]$, then*

*(a) $DZ_x^j(w)[\Delta] = \sum_{i \in [n_{\text{in}}]} U_x^i \Delta^{i,j} \in \mathbb{R}^{d_{\text{out}}}$;*

*(b) $\{DZ_x^{j*}(w)[\tau^j]\}^{i,j} = [U_x^i]^* \tau^j \in \mathbb{R}^{d_k}$;*

*(c) $DZ_x^j(w) \circ DZ_x^{j*}(w)[\tau^j] = \sum_{i \in [n_{\text{in}}]} U_x^i [U_x^i]^{j*} \tau^j \in \mathbb{R}^{d_{\text{out}}}$.*

Since the elements of $U_x^i$ are the values of $x$, the identity in (6) and Lemma 3.1(b) imply that the squared norm of $\nabla_w \phi_x^j(w,b)$ can be expressed solely in terms of $x$ and the downstream gradient $g^j$ in (5). In the next two subsections, we give two different expressions for $\|\nabla_w \phi_x^j(w,b)\|$ and present their corresponding algorithms.

### 3.1 MEMORY-EFFICIENT NORM COMPUTATION

This subsection presents two in-place algorithms for computing the desired squared gradient norm.

We first present a "direct" expression for $\nabla_w \phi_x^j(w,b)$ in terms of $x$ and $g^j$ using (6). The proof is postponed to Appendix A.

**Lemma 3.2.** *Let $g^j \in \mathbb{R}^{n_{\text{out}} \times d_{\text{out}}}$ be as in (6) and $s \geq 1$ be given. Then, it holds that the value of the gradient $\nabla_w \phi_x^j(w, b)$ at the $i$-th input channel, $j$-th output channel, and $m$-th output dimension is given by*

$$[\nabla_w \phi_x^j(w, b)]_m^{i,j} = \sum_{\ell \in [d_{\text{out}}]} (x_{[\ell-1]s+m}^i)(g_\ell^j). \tag{7}$$

The above result shows that when we are given $x$ and $g$, we can compute $\|\nabla_w \phi_x(w, b)\|^2$ by performing a sequence of in-place operations. For ease of reference, we present one variant of these operations in Algorithm 3.1, which can be viewed as an in-place modification of the FastGradClip algorithm in Lee & Kifer (2021b). It is immediate that Algorithm 3.1 requires

$$T_{\text{direct}} := n_{\text{in}} n_{\text{out}} d_k d_{\text{out}} \tag{8}$$

floating-point operations (FLOPS), but only $O(1)$ additional storage.

---

**Algorithm 3.1** Direct squared norm computation with in-place operations

---

1: *Input*: stride length $s \geq 1$, layer input $x \in \mathbb{R}^{n_{\text{in}} \times d_{\text{in}}}$, and gradient $g \in \mathbb{R}^{n_{\text{out}} \times d_{\text{out}}}$;
2: *Output*: value of $\|\nabla_w \phi_x(w, b)\|^2$;
3: Define $\mathcal{J}_m := \{([\ell-1]s + m, \ell) : \ell \in [d_{\text{out}}]\}$ for $m \in [d_k]$

4: **return** $\sum_{i \in [n_{\text{in}}]} \sum_{j \in [n_{\text{out}}]} \sum_{m \in [d_k]} \left( \sum_{(p,q) \in \mathcal{J}_m} x_p^i g_q^j \right)^2$

---

We now present a special expression for $\|\nabla_w \phi_x(w, b)\|^2$ that is reminiscent of a similar expression in the "Ghost Clipping" algorithm from Bu et al. (2022). The proof is postponed to Appendix A.

**Lemma 3.3.** *Let $g^j \in \mathbb{R}^{d_{\text{out}}}$ be as in (6), let $s \geq 1$ be given, and define*

$$X_{\ell,\ell'} := \sum_{i \in [n_{\text{in}}]} \sum_{m \in [d_k]} (x_{[\ell-1]s+m}^i)(x_{[\ell'-1]s+m}^i), \quad G_{\ell,\ell'} := \sum_{j \in [n_{\text{out}}]} g_\ell^j g_{\ell'}^j$$

*where $\ell, \ell' \in [d_{\text{out}}]$ are indices over the output dimension. Then, it holds that*

$$\|\nabla_w \phi_x(w, b)\|^2 = \sum_{j \in [n_{\text{out}}]} \langle \mathcal{A}_x \mathcal{A}_x^*, [g^j][g^j]^* \rangle = 2 \sum_{1 \leq \ell < \ell' \leq d_{\text{out}}} X_{\ell,\ell'} G_{\ell,\ell'} + \sum_{\ell \in [d_{\text{out}}]} X_{\ell,\ell} G_{\ell,\ell}, \tag{9}$$

*where $\mathcal{A}_x$ is the matrix in $\mathbb{R}^{d_{\text{out}} \times d_k}$ corresponding the operator of the same name in (5).*

Similar to Lemma 3.2, the above result also yields a sequence of in-place operations for computing $\|\nabla_w \phi_x(w, b)\|^2$. As before, for ease of reference, we present one variant of these operations in Algorithm 3.2. It is straightforward to see that, for a fixed outer index pair $(\ell, \ell')$ in the expression for $P$, the computation of the inner sum involving $x$ (resp. $g$) requires $n_{\text{in}} d_k$ FLOPS (resp. $n_{\text{out}}$). Consequently, computing $P$ and $Q$ in Algorithm 3.2 requires

$$T_{\text{ghost}} := \left[ d_{\text{out}} + \frac{d_{\text{out}}(d_{\text{out}} - 1)}{2} \right] (n_{\text{in}} d_k + n_{\text{out}}) = \Theta(d_{\text{out}}^2 [n_{\text{in}} d_k + n_{\text{out}}]) \tag{10}$$

total FLOPS but also only $O(1)$ additinal storage.

---

**Algorithm 3.2** Ghost Clipping-based squared norm computation with in-place operations

---

1: *Input*: layer input $x \in \mathbb{R}^{n_{\text{in}} \times d_{\text{in}}}$ and gradient $g \in \mathbb{R}^{n_{\text{out}} \times d_{\text{out}}}$;
2: *Output*: value of $\|\nabla_w \phi_x(w, b)\|^2$;
3: Define $\mathcal{J}_{\ell,\ell'} := \{([\ell-1]s + m, [\ell'-1]s + m) : m \in [d_k]\}$ for $\ell, \ell' \in [d_{\text{out}}]$
4: Compute $P \leftarrow \sum_{1 \leq \ell < \ell' \leq d_{\text{out}}} \left( \sum_{i \in [n_{\text{in}}]} \sum_{(p,q) \in \mathcal{J}_{\ell,\ell'}} x_p^i x_q^i \right) \left( \sum_{j \in [n_{\text{out}}]} g_\ell^j g_{\ell'}^j \right)$

5: Compute $Q \leftarrow \sum_{\ell \in [d_{\text{out}}]} \left( \sum_{i \in [n_{\text{in}}]} \sum_{(p,q) \in \mathcal{J}_{\ell,\ell}} x_p^i x_q^i \right) \left( \sum_{j \in [n_{\text{out}}]} g_\ell^j g_\ell^j \right)$
6: **return** $2P + Q$

---

## 3.2 Fourier-based norm computation

This subsection presents an algorithm based on the discrete Fourier transform (DFT) for computing the desired squared gradient norm.

We first define $\mathrm{rev} : \mathbb{R}^n \mapsto \mathbb{R}^n$ (resp. $\mathrm{diag} : \mathbb{R}^n \mapsto \mathbb{R}^{n \times n}$) to be the linear operator that reverses the order of its input (resp. diagonalizes its input). Explicitly, these operators are given by

$$\mathrm{rev}([x_1, x_2, \ldots, x_n]) = [x_n, \ldots, x_2, x_1], \quad [\mathrm{diag}(x)]_{i,j} = \begin{cases} x_i, & \text{if } i = j \\ 0, & \text{otherwise} \end{cases}, \quad \forall i, j \in [n], \quad (11)$$

for every $x \in \mathbb{R}^n$. Now, let us recall the notion of a circulant matrix and its relationship to the DFT. A circulant matrix $C \in \mathbb{R}^{n \times n}$ (resp. an anti-circulant matrix $\zeta \in \mathbb{R}^{n \times n}$) is a Toeplitz (resp. anti-Toeplitz) matrix of the form

$$C = \begin{bmatrix} c_0 & c_{n-1} & \cdots & c_1 \\ c_1 & c_0 & \cdots & c_2 \\ \vdots & \vdots & \ddots & \vdots \\ c_{n-1} & c_{n-2} & \cdots & c_0 \end{bmatrix}, \quad \zeta = \begin{bmatrix} c_1 & \cdots & c_{n-1} & c_0 \\ c_2 & \cdots & c_0 & c_1 \\ \vdots & \cdot^{\cdot^{\cdot}} & \vdots & \vdots \\ c_0 & \cdots & c_{n-2} & c_{n-1} \end{bmatrix}, \quad (12)$$

for some $c \in \mathbb{R}^n$. Notice that consecutive rows of a circulant (resp. anti-circulant) matrix contain the same entries of $c$ but are cyclically shifted from left to right (resp. right to left).

The next result relates circulant matrices in $\mathbb{R}^{n \times n}$ with the $n$-th order DFT, and its proof can be found, for example, in (Gray et al., 2006).

**Lemma 3.4.** *If $C \in \mathbb{R}^{n \times n}$ is a circulant matrix and $c$ is its first column, then $C = \mathcal{F}_n^{-1} \mathrm{diag}(\mathcal{F}_n c) \mathcal{F}_n$, where $\mathcal{F}_n$ is the $n$-th order DFT.*

Using the above result, it is straightforward to see that if $\zeta \in \mathbb{R}^{n \times n}$ is an anti-circulant matrix whose first row is $\mathrm{rev}(c)$

$$\zeta \tau = \mathrm{rev}(\mathcal{F}_n^{-1} \mathrm{diag}[\mathcal{F}_n \mathrm{rev}(c)] \mathcal{F}_n \tau), \quad \forall \tau \in \mathbb{R}^n. \quad (13)$$

Returning to our main goal, the primary insight of this section is that we can express $\nabla_w \phi_x^j(w, b)$ (and, consequently, $\nabla_w \phi_x(w, b)$) as an application of an anti-circulant matrix with some auxiliary (but simple) linear transforms. The details of this perspective, and its computational implications, are given in the following result, whose proof is postponed to Appendix A.

**Proposition 3.5.** *Let $\zeta_x^i \in \mathbb{R}^{d_{\mathrm{in}} \times d_{\mathrm{in}}}$ denote the anti-circulant matrix whose first row is $x^i$. Moreover, define the block matrices $Q \in \mathbb{R}^{d_{\mathrm{in}} \times d_k}$ and $R \in \mathbb{R}^{d_{\mathrm{out}} \times d_{\mathrm{in}}}$ by*

$$Q := \begin{bmatrix} I_{d_k} \\ 0_{(d_{\mathrm{in}} - d_k) \times d_k} \end{bmatrix}, \quad [R]_{n,m} = \begin{cases} 1, & \text{if } m = s(n-1) + 1 \\ 0, & \text{otherwise}, \end{cases} \quad \forall (n, m) \in [d_{\mathrm{in}}] \times [d_{\mathrm{out}}], \quad (14)$$

*where $I_n$ (resp. $0_{n \times m}$) denotes the identity matrix in $\mathbb{R}^{n \times n}$ (resp. zero matrix in $\mathbb{R}^{n \times m}$). Then, it holds that*

*(a) for every $i \in [n_{\mathrm{in}}]$, we have $U_x^i = R \zeta_x^i Q$;*

*(b) if $g^j \in \mathbb{R}^{d_{\mathrm{out}}}$ is as in (5), then*

$$\left[ \nabla_w \phi_x^j(w, b) \right]^i = Q^* \circ \mathrm{rev} \circ \mathcal{F}_{d_{\mathrm{in}}}^{-1} \left( [\mathcal{F}_{d_{\mathrm{in}}} \circ \mathrm{rev}(x^i)] \odot [\mathcal{F}_{d_{\mathrm{in}}} R^* g^j] \right) \quad \forall i \in [n_{\mathrm{in}}], \quad (15)$$

*where $\odot$ denotes the Hadamard product.*

Before proceeding, let us give a few remarks. First, for $y \in \mathbb{R}^{d_{\mathrm{in}}}$ and $z \in \mathbb{R}^{d_{\mathrm{out}}}$, we have that $Q^* y$ returns the first $d_k$ rows of $y$ and $R^* z$ returns a padded version of $z$ in which $[R^* z]_{s(i-1)+1} = z_i$ for $i \in [d_{\mathrm{out}}]$ and $[R^* z]_j$ is zero at all other indices $j$. Second, in view of the first remark, we have that for any $y \in \mathbb{R}^{d_{\mathrm{in}}}$, both of the quantities $(Q^* \circ \mathrm{rev})(y)$ and $R^* g^j$ can be computed using $\Theta(d_{\mathrm{in}})$ FLOPS.

We now present a general algorithm in Algorithm 3.3 that leverages (15) to calculate $\|\nabla_w \phi_x(w, b)\|^2$. Notice, in particular, that it can be specialized to different choices of the DFT oracle $\mathcal{F}_{d_{\mathrm{in}}}$.

---

**Algorithm 3.3** DFT-based squared norm computation

---

1: *Input*: layer input $x \in \mathbb{R}^{n_{\text{in}} \times d_{\text{in}}}$, gradient $g \in \mathbb{R}^{n_{\text{out}} \times d_{\text{out}}}$, and oracle $\mathcal{F}_{d_{\text{in}}}$ that performs the $(d_{\text{in}})$-th order DFT;
2: *Output*: value of $\|\nabla_w \phi_x(w, b)\|^2$;
3: Define $\text{rev}(\cdot)$ and $(Q, R)$ to be as in (11) and (14), respectively
4: **for** $i, j \in [n_{\text{in}}] \times [n_{\text{out}}]$ **do**
5: $\quad v^{i,j} \leftarrow Q^* \circ \text{rev} \circ \mathcal{F}_{d_{\text{in}}}^{-1}([\mathcal{F}_{d_{\text{in}}} \circ \text{rev}(x^i)] \odot [\mathcal{F}_{d_{\text{in}}} R^* g^j])$
6: $\quad r^{i,j} \leftarrow \sum_{\ell=1}^{d_k} (v_\ell^{i,j})^2$
7: **end for**
8: **return** $\sum_{(i,j) \in [n_{\text{in}}] \times [n_{\text{out}}]} r^{i,j}$

---

The next result presents the runtime and storage complexity of a specialization of Algorithm 3.3, where we use a fast (discrete) Fourier transform oracle. Specifically, it is well-known that DFT can be implemented in time $O(d \log d)$ using the FFT method (Duhamel & Vetterli, 1990). The proof can be found in Appendix A.

**Theorem 3.6.** *Let $\bar{\mathcal{F}}_{d_{\text{in}}}$ be an FFT oracle which, for any $v \in \mathbb{R}^{d_{\text{in}}}$, computes $\bar{\mathcal{F}}_{d_{\text{in}}} v$ in $\Theta(d_{\text{in}} \log d_{\text{in}})$ FLOPS. Then, there is an implementation of Algorithm 3.3 with $\mathcal{F}_{d_{\text{in}}} = \bar{\mathcal{F}}_{d_{\text{in}}}$ that consumes at most*

$$T_{\text{fft}} = \Theta(n_{\text{in}} n_{\text{out}} d_{\text{in}} \log d_{\text{in}}) \tag{16}$$

*total FLOPS and $\Theta(d_{\text{in}})$ additional storage.*

## 3.3 TECHNICAL DISCUSSION

This subsection discusses two topics, namely, (i) how the runtime and storage costs from the previous subsections compare under different settings and (ii) how our results generalize to higher-dimensional inputs.

We start by comparing how the runtime complexities $T_{\text{direct}}$, $T_{\text{ghost}}$, and $T_{\text{fft}}$ in (8), (10), and (16), respectively. For simplicity, let us assume that the stride length is $s = 1$ and let $d \geq 1$ and $n \geq 1$ be arbitrary. First, when $d_{\text{in}}, d_{\text{out}}, d_k = \Theta(d)$ and $n_{\text{in}}, n_{\text{out}} = O(1)$, we have that

$$T_{\text{fft}} = \Theta(d \log d) \preceq T_{\text{direct}} = \Theta(d^2) \preceq T_{\text{ghost}} = \Theta(d^3).$$

where $A \preceq B$ means that $A$ is asymptotically more efficient than $B$ in terms of runtime. Second, when $d_{\text{in}}, d_{\text{out}} = \Theta(d)$, $d_k = O(1)$, and $n_{\text{in}}, n_{\text{out}} = \Theta(1)$, we have that

$$T_{\text{direct}} = \Theta(d) \preceq T_{\text{fft}} = \Theta(d \log d) \preceq T_{\text{ghost}} = \Theta(d^2).$$

Finally, when $d_{\text{in}}, d_k = \Theta(d)$, $d_{\text{out}} = O(1)$, $n_{\text{in}}, n_{\text{out}} = \Theta(n)$, we have that

$$T_{\text{ghost}} = \Theta(nd) \preceq T_{\text{direct}} = \Theta(n^2 d) \preceq T_{\text{fft}} = \Theta(n^2 d \log d).$$

From the above comparisons, we can see that each of our proposed methods outperforms the others in certain regimes, so there is not a method that performs universally better across different choices of $n_{\text{in}}, n_{\text{out}}, d_k, d_{\text{in}}$, and $d_{\text{out}}$.

We next remark that our results are formally presented for the case of multiple input-output channels and one-dimensional (per-example) inputs. When $x$ is an $d$-dimensional input, it is straightforward to develop analogous version for Algorithms 3.1–3.2. However, the analogous version of Algorithm 3.3 requires more care. In particular, we would need to develop higher-order versions of (12), replace the one-dimensional Fourier transform in Algorithm 3.3 with its $d$-dimensional variant, and replace the operators $(Q, R)$ in Algorithm 3.3 with higher-order variants.

In the special case of the two-dimensional DFT, which is useful when the inputs are images, it is known (Azimi-Sadjadi & King, 1987) that a version of Lemma 3.4 holds where $C$ is replaced by a block-circulant matrix, i.e., where each $c_i$ in (12) is replaced by a matrix. Consequently, the version of Algorithm 3.3 for a two-dimensional (per-example) input array $x$ directly follows from this result by replacing (i) $\mathcal{F}_{d_{\text{in}}}$ by its analogous two-dimensional DFT, (ii) $\text{rev}(\cdot)$ by the operator that reverses a two-dimensional input array lexicographically, and (iii) $Q$ and $R$ by their block two-dimensional variants. We posit that the $d$-dimensional version of Algorithm 3.3 is one where changes (i)–(iii) are applied in the $d$-dimensional setting, i.e., with blocks of $d$-dimensional arrays instead of two-dimensional matrices.

## 4 NUMERICAL RESULTS

In this section we perform an empirical evaluation of the algorithms we proposed in our work and compare their performance to the algorithms proposed by Bu et al. (2022). Since the only part of the DP-SGD training pipeline we modified is the norm computation step, to make the comparison across our algorithms and the algorithms from prior work (Bu et al., 2022) more precise, this section is devoted solely to this part of the training; in other words, we do not perform full training of CNNs using DP-SGD, but we rather evaluate the gradient norm computation functions for various types of layers and input examples. The settings we consider are motivated by our discussion in Section 3.3. In particular, we focus on three different regimes. Firstly, we consider families of layers where the input, output dimensions and kernel size scale as $\Theta(d)$, and the number of input, output channels are small constants Subsequently, we focus on families of layers where the input, output dimensions scale as $\Theta(d)$, the kernel size is $\Theta(1)$, and the number of input, output channels is also $\Theta(1)$. Finally, we consider the setting where the input, outputs dimensions, and kernel size scale are $\Theta(1)$, and the number of input, output channels scale as $\Theta(n)$. For all the experiments, we initialize random inputs to a particular layer of the CNN that is configured according to the underlying setting, and for every configuration we repeat the experiment 5 times to reduce the variance. Moreover, to give a more detailed comparison of the difference of the running time and memory consumption the $y$-axis of these plots is in $\log$-scale. In all the experiments the stride of the kernel is 1. All the experiments were executed in Python on a standard laptop.

**Setting #1:** $d_{\text{in}}, d_{\text{out}}, d_k = \Theta(d), n_{\text{in}}, n_{\text{out}} = O(1)$. As we alluded to before, we first consider the setting where the input dimension, output dimension and kernel size are of the same order of magnitude and they are all much larger than number of input, output channels. The runtime and memory consumption comparisons are depicted in Figure 4.1. For this setting, our in-place ghost-norm algorithm was performing significantly worse than the rest of the methods and was increasing the computation time of the experiment significantly, so we have not displayed it. We can see that as $d$ increases, the advantage of the FFT-based method becomes increasingly more significant. Moving on to the comparison of the memory consumption, this experiment illustrates that our approaches have significantly lower memory requirements than those from prior work, as the memory consumption remains constant as $d$ increases.

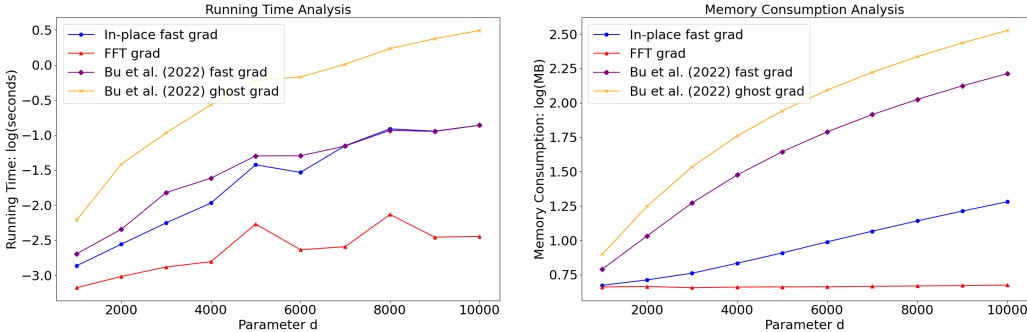

Figure 4.1: Runtime and peak memory consumption: $d_{\text{in}} = d, d_k, d_{\text{out}} = d/2, n_{\text{in}}, n_{\text{out}} = 5$

**Setting #2:** $d_{\text{in}}, d_{\text{out}} = \Theta(d), d_k = O(1), n_{\text{in}}, n_{\text{out}} = O(1)$. Next, we consider the setting where the input, output dimension scale as $\Theta(d)$, and the kernel size, input channels, and output channels are small constants. The runtime, memory consumption comparison are depicted in Figure 4.2. For this setting, our in-place direct norm computation algorithm outperforms all other methods, verifying our theoretical analysis. Turning our attention to the memory comparison, we observe that the ghost-norm computation algorithm from Bu et al. (2022) has the largest memory requirement, whereas all three of our algorithms require the least amount of memory.

**Setting #3:** $d_{\text{in}}, d_{\text{out}}, d_k = O(1), n_{\text{in}}, n_{\text{out}} = \Theta(n)$. Finally, we consider the setting where the input dimension, output dimension, and kernel size are small constants and we let the number of input, output channels be $\Theta(n)$. The runtime, memory consumption comparison are depicted in Figure 4.3.

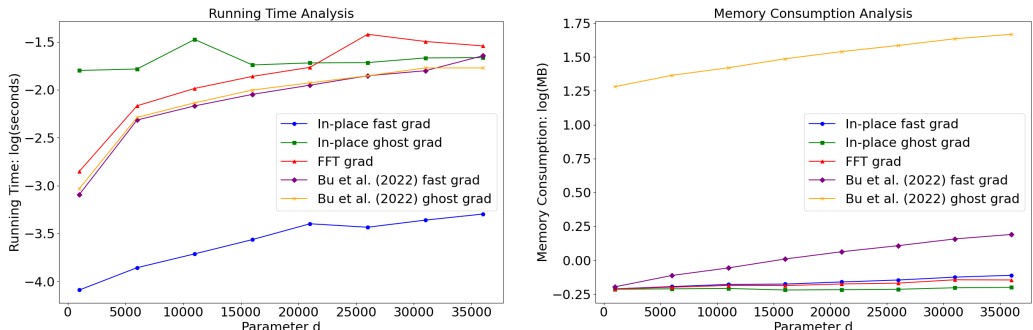

Figure 4.2: Runtime and peak memory consumption: $d_{\text{in}} = d, d_k = d - 14, d_{\text{out}} = 13, n_{\text{in}}, n_{\text{out}} = 3$

In this setting, we observe that for large enough $n$ our in-place ghost-norm computation and the ghost-norm computation of Bu et al. (2022) outperform all other methods, verifying our theoretical analysis. Interestingly, the direct gradient computation method of Bu et al. (2022) is also performing very well; this is due to Python implementation features, since matrix multiplication, which is used for the direct gradient computation of Bu et al. (2022) speeds up the computation. Moreover, we observe that our approach is significantly more memory-efficient than Bu et al. (2022).

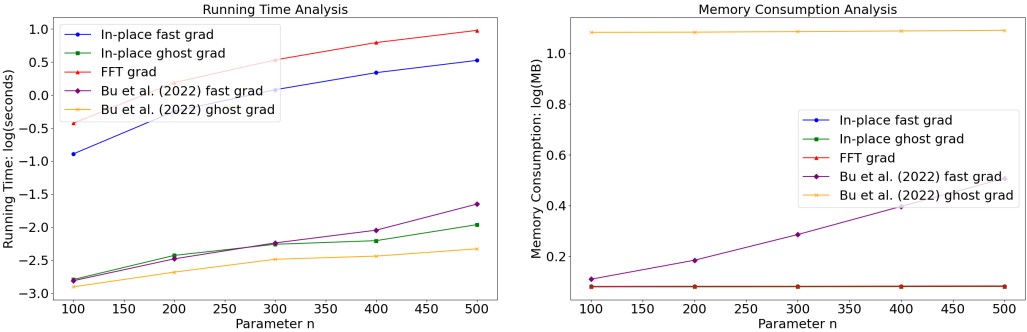

Figure 4.3: Runtime and peak memory consumption: $d_{\text{in}} = 10, d_k = 10, d_{\text{out}} = 1, n_{\text{in}}, n_{\text{out}} = n$

## 5 CONCLUSION

In this work we have proposed three new methods for gradient norm computation which can significantly improve the runtime and memory efficiency of DP-SGD over prior work for certain architectures of CNNs. We have rigorously analyzed the theoretical improvements of our algorithms, which are also supported by numerical experiments.

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

## A  TECHNICAL PROOFS

This appendix gives the proofs of the manuscript's main results.

*Proof of Lemma 3.1.* For simplicity, denote $U^i = U^i_x$ and $\mathcal{D}_x := DZ_x(w)$.

(a) This is immediate from the linearity of $\mathcal{D}_x$.

(b) From part (a) and the definition of the adjoint, we have

$$\langle \mathcal{D}_x \Delta, \tau \rangle = \sum_{i \in [n_{in}]} \langle \tau, U^i \Delta^{i,j} \rangle = \sum_{i \in [n_{in}]} \langle [U^i]^* \tau, \Delta^{i,j} \rangle = \langle \mathcal{D}_x^* \tau, \Delta \rangle .$$

(c) Using parts (a) and (b), we have $\mathcal{D}_x \mathcal{D}_x^* \tau = \sum_{i \in [n_{in}]} U^i [\mathcal{D}_x^* \tau]^{i,j} = \sum_{i \in [n_{in}]} U^i [U^i]^* \tau.$ □

*Proof of Lemma 3.2.* In view of Lemma 3.1(b) and (6) it suffices to show that the $m$-th column of $U^i_x$ is the vector $\mathrm{Col}^i_m := [x^i_m, x^i_{s+m}, \ldots, x^i_{(d_{out}-1)s+m}] \in \mathbb{R}^{d_{out}}$. Indeed, recall that the $\ell$-th row of $U^i_x$ is the $\ell$-th window of the input $x$ and is given by $\mathrm{Row}^i_k := [x^i_{(\ell-1)s+1}, \ldots, x^i_{(\ell-1)s+d_k}] \in \mathbb{R}^{d_k}$. Fixing a column index $m$, it is clear that the values in the $m$-th index of $\mathrm{Col}^i_k$ for $k \in [d_k]$ form the elements of $\mathrm{Col}^i_m$. □

*Proof of Lemma 3.3.* The first identity in (9) is immediate from (6) and the definition of the adjoint of a linear operator. For the second identity, note that (4), (5), and Lemma 3.1(c) imply that $\mathcal{A}_x = \sum_{i \in [n_{in}]} U^i_x [U^i_x]^*$. Hence, in view of the definition of $X_{\ell,\ell'}$ and $G_{\ell,\ell'}$, it suffices to show that the entry in the $\ell$-th row and $\ell'$-th column of $U^i_x [U^i_x]^*$ is given by

$$\left[ U^i_x \{U^i_x\}^* \right]_{\ell,\ell'} = \sum_{m \in [d_k]} (x^i_{[\ell-1]s+m})(x^i_{[\ell'-1]s+m}).$$

Indeed, recall that the $k$-th row of $U^i_x$, say $\mathrm{Row}^i_k$, contains the $\ell$-th window of the input array $x$. For a given stride $s$ and kernel size $d_k$, clearly we have $\mathrm{Row}^i_k = [x^i_{(\ell-1)s+1}, \ldots, x^i_{(\ell-1)s+d_k}]$. □

*Proof of Proposition 3.5.* (a) Observe that for any matrix $M \in \mathbb{R}^{d_{in} \times d_{in}}$, we have that $MQ$ returns the first $d_k$ columns of $M$ and $RM$ returns rows $1, s+1, \ldots, d_{out} - 1 + s$ of $M$. The conclusion now follows from the previous observation and the fact that the rows of $U^i_x$ contain the windows of $x$ of size $d_k$ and stride $s$.

(b) Using part (a) and (13) with $\zeta = \zeta^i_x$, we have that, for any $\tau \in \mathbb{R}^{d_k}$,

$$[U^i_x]^* \tau = Q^* [\zeta^i_x]^* R^* \tau = Q^* \circ \mathrm{rev} \left( \mathcal{F}^{-1}_{d_{in}} \mathrm{diag} \left[ \mathcal{F}_{d_{in}} \mathrm{rev}(x^i) \right] \mathcal{F}_{d_{in}} R^* \tau \right)$$

$$= Q^* \circ \mathrm{rev} \circ \mathcal{F}^{-1}_{d_{in}} \left( \left[ \mathcal{F}_{d_{in}} \mathrm{rev}(x^i) \right] \odot \left[ \mathcal{F}_{d_{in}} R^* \tau \right] \right)$$

Consequently, using the above identity with $\tau = g^j$ and Lemma 3.1(b) we have that

$$[\nabla_w \phi^j_x(w,b)]^i = [U^i_x]^* g^j = Q^* \circ \mathrm{rev} \circ \mathcal{F}^{-1}_{d_{in}} \left( \left[ \mathcal{F}_{d_{in}} \mathrm{rev}(x^i) \right] \odot \left[ \mathcal{F}_{d_{in}} R^* g^j \right] \right) .$$

□

*Proof of Theorem 3.6.* It suffices to describe the costs of computing $v^{i,j}$ and $r^{i,j}$ for $i \in [n_{\text{in}}]$ and $j \in [n_{\text{out}}]$.

For fixed $(i, j)$, computing $v^{i,j}$ can be done by: (i) computing $a = R^* g^j$ in a $\Theta(d_{\text{in}})$ runtime and storage cost (see the remarks following Proposition 3.5), (ii) computing $\hat{a} = \mathcal{F}_{d_{\text{in}}} a$ and $\hat{c} = \mathcal{F}_{d_{\text{in}}} \circ \text{rev}(x^i)$ in a $\Theta(d_{\text{in}} \log d_{\text{in}})$ runtime cost, (iii) computing $\hat{e} = \hat{c} \odot \hat{a}$ in a $\Theta(d_{\text{in}})$ runtime and storage cost, (iv) computing $e = \mathcal{F}_{d_{\text{in}}}^{-1} \hat{e}$ in a $\Theta(d_{\text{in}} \log d_{\text{in}})$ runtime cost, and (v) computing $Q^* \circ \text{rev}(e)$ in a $\Theta(d_{\text{in}})$ runtime and storage cost (see the remarks following Proposition 3.5). Summing the previous terms results in a $\Theta(d_{\text{in}} \log d_{\text{in}})$ runtime cost and $\Theta(d_{\text{in}})$ storage cost. For fixed $(i, j)$, computing $r^{i,j}$, given $v^{i,j}$, can be done by an accumulating sum in a runtime and storage cost of $\Theta(d_k)$ and $O(1)$, respectively.

Summing all the above costs over $i \in [n_{\text{in}}]$ and $j \in [n_{\text{out}}]$ (new temporary variables for the computations of $v^{i,j}$ and $r^{i,j}$) yields a storage cost of $\Theta(d_{\text{in}})$ and a runtime cost of $\Theta(T)$, where

$$T = n_{\text{in}} n_{\text{out}} \left( \underbrace{d_{\text{in}} \log d_{\text{in}}}_{v^{i,j}} + \underbrace{d_k}_{r^{i,j}} \right) = \Theta(n_{\text{in}} n_{\text{out}} [d_{\text{in}} \log d_{\text{in}}]),$$

where the last identity follows from the fact that $d_k \leq d_{\text{in}}$. $\quad\square$

