# OpenReview forum: "Efficient Gradient Clipping Methods in DP-SGD for Convolution Models"
_ICLR.cc/2025/Conference — Submitted to ICLR 2025_

### Official Review · Reviewer_rBnc · 2024-10-28

**Soundness:** 3
**Presentation:** 3
**Contribution:** 2
**Rating:** 5
**Confidence:** 3

**Summary:**

This paper presents and studies memory and compute efficient methods for computing per-instance gradient norms in the context of training convolutional layers with DP-SGD. Three different methods are analyzed and studied. The first two methods are fairly simple variations to well-known methods (resp. Im2col followed by GEMM and ghost-clipping) by making them in-place; the third method exploits the connection between convolutions and Fourier transforms to transform the convolution operation to point-wise products in frequency space. The efficiency of the proposed methods are benchmarked against previous state of art methods in some-what artificial settings.

**Strengths:**

1. Applying FFT to accelerate per-instance gradient computation for convolution layers is new to my best knowledge.

2. The mathematical formulation in the background section, in particular the use of operator notation to show that the gradient of convolution with respect to the kernel is the transposed convolution between downstream gradient and input, is neat.

**Weaknesses:**

1. My primary concern is whether the computation of per-instance gradient norms warrants treatment as a research question, as opposed to being a matter of implementation. I admit that it may not be fair to ask this particular paper of this question, as I am aware that there are multiple published works tackling this exact problem.
These works point to most ML frameworks (e.g. pytorch, tensorflow) returning only the aggregated per-batch parameter gradient as a fundamental roadblock to computing per-instance gradient norms, but this is only true if we restrict ourselves to the python API implemented by these high-level frameworks.
As such, I have yet to encounter a convincing argument as to why this per-instance gradient norm cannot be trivially obtained by computing it at the kernel level, right after per-instance gradients have been computed, before they are reduced into per-batch gradients.
I acknowledge that there are nuances as convolutions and their backward computations are often divided and parallelized into per-patch operations, yet the general spirit remains true that the fine-grained (whether per-instance, per input channel, or even per sliding window) gradients are always computed first before they are aggregated via reduce operations, and that careful book-keeping of their norms can provide us the desired per-instance gradient norm with minimal compute and memory overhead.

2. Measuring memory consumption in a memory-managed language like python is rather unreliable as the memory usage behavior can differ significantly depending on implementation and how memory usage is measured. More details, including source code, is required to make a convincing empirical argument as to the superior memory efficiency of the proposed method.

3. The overall impact of the methods presented in this paper seems limited: method 1 and method 2 are mostly re-hashes of the direct im2col+GEMM method and the ghost-clipping method with minute differences, and the FFT based method, while more novel, is the least performant out of the three in realistic settings where the kernel size is small and the number of channels is high.

**Questions:**

Please address point 1 in the weaknesses section.

What is $l_x$ in (4)?

---

> ### Author Response · Authors · 2024-11-21
>
> We would like to thank the reviewer for their review. Please find our answers to your questions below.
>
> Weakness 1:  Thank you for raising this point. Improving the runtime and memory complexity of the clipping step of DP-SGD merits attention and is not merely a matter of the underlying implementation. Typically, after the back-propagation step, the learner has access to the partial derivative of the output, denoted as $g$ in our algorithms. The “direct” approach you allude to, whose in-place implementation we provide in Algorithm 3.1, can in some cases be significantly slower than the other two methods we propose in this paper. This happens even if the batch size is 1, so the size of the batches is not the only bottleneck for the performance overhead., Several works discuss the difficulties of using DP algorithms in practical applications mention that the running time of clipping is one of the major issues and blockers to deploying machine learning models trained on sensitive data, e.g., in medical fields (see, for example https://arxiv.org/pdf/2303.00654, Section 4.5, in particular the discussion about slower training).
>
> Weakness 2:
> We were measuring the peak memory of the operations. Moreover, our theoretical analysis of the algorithms proves that the in-place algorithms have less memory overhead than prior work. We will release the code in the next version of our work.
>
> Weakness 3:
> Even though the first two methods appear to be similar to prior work, we followed a different conceptual path to obtain them. Prior work considered “unrolling” the gradients to large matrices, and performing matrix multiplication, which required a large amount of additional memory and computation. On the other hand, our approach is to analyze the updates as abstract linear operators and, consequently, to develop a more efficient scheme for evaluating them. We acknowledge the limitations of the FFT method with small kernels, but disagree that this makes it impractical. Large kernels have recently gained attention, particularly in applications involving long-context inputs [https://arxiv.org/pdf/2203.06717] and could further have an impact with language modeling for capturing complex long-range dependencies. Our bounds and experiments demonstrate that in these specific scenarios our FFT approach exhibits superior performance compared to baselines. This highlights the practicality and relevance of this approach, in addition to its novelty.
>
> $\ell_x$ in 4 denotes the loss function for a fixed $x$.

---

> > ### Comment · Reviewer_rBnc · 2024-11-24
> > **Rebuttal response from reviewer**
> >
> > I thank the authors for the response. I can mostly agree with the author's rebuttal regarding points 2 and 3 and have adjusted my score accordingly, but I would like to further discuss point 1.
> >
> > I think the direct approach I referred to in the review is different from that of algorithm 3.1, but instead requires lower level access to the returns of the cuda kernel (or its analogous abstraction in TPUs) operation before the reduction by the cpu process. To leverage data parallism, most cuda convolution implementations are first sliced at the batch dimension, and then at the sliding window dimension, before being distributed across GPU threads. Hence, there is a program state in the natural computation of the parameter gradient where many threads have each computed the partial gradient of the parameter with respect to one example, from which the per-example gradient norm can be computed and returned from the thread.
> >
> > For a less bare-metal example, consider instead a distributed training setup where each GPU device is training at batch size one (many modern models are trained at very small per device batch sizes across thousands of devices, hence I wouldn't consider this a far-fetched setting). At the end of each backward pass on each device, the per-example gradient is computed and stored in the process on each device. In non-DP training, an AllReduce communication primitive is then invoked to sum the gradient across all devices, and divided by the number of devices to obtain the average gradient. For DP training, we can append the gradient norm computation in each process from the already available per-example gradient (which has additional runtime complexity $O(n_{in} n_{out} d_{k})$ and memory $O(1)$ per device), and then use an AllGather primitive to distribute the per-example norm for computing the correct noising amplitude.
> >
> > It appears to me that the proposed algorithms would have worse runtime complexity and memory footprint than this direct approach, at least in the distributed setting. Yet, at the bare-metal level, all tensor operations are distributed across cuda kernels, which makes believe that a direct approach would be feasible and performant, if implemented carefully.

---

> > > ### Author Response · Authors · 2024-11-25
> > >
> > > Thank you for the clarification. We agree that, under access to thousands of accelerators, the parallelized SPMD approach you described may be more performant and discussing parallelization is more subtle than what we originally discussed. We will be adding more details to an updated version.
> > >
> > > Please find below our analysis on how our approach would fare when we consider the effects of parallelization.
> > >
> > > While parallelization can be beneficial in the above setting, in more constrained settings -- where the number of accelerators is limited and the underlying model is large (>10 billion parameters) -- most implementations of DP-SGD operate serially (typically by accumulating clipped gradients in a buffer) or with limited parallelism (e.g., parallelizing over small subsets of the batch and sliding window axes).
> > >
> > > In this constrained setting, memory savings are much more important. In the scheme we are proposing for Algorithm 3.1, we remove the sliding window axis as a possible axis for sharding/parallelization to potentially save on memory costs. More specifically, each accelerator would need access to the entire input for each example in a (sharded) batch.
> > >
> > > For additional clarity, let's consider a concrete example. Suppose:
> > >
> > > * The number of accelerators is $A$.
> > > * Batch size is $b$.
> > > * Inputs are 2D images of size $n-1$-by-$n-1$ for even $n$
> > > * The kernel is a 2D window of size $(n/2)$-by-$(n/2)$
> > > * The kernel slides across the images one-block at a time
> > >
> > > It is straightforward to show that the number of elements formed by the "unrolled" input is $M := (n/2)^4 = n^4 / 16$ and the smallest data shard that can be passed to an accelerator is is $(n/2)^2$ (corresponding to one window of the input). Here, $M$ is the dimension of the sliding window axis.
> > >
> > > In your proposed SPMD setup, where we can slice across both the batch and sliding window axis, we would need to distribute $\Theta(bn^4)$ pieces of data across $A$ accelerators, where each data shard must be at least size $\Theta(n^2)$.
> > >
> > > In our setup, where we can only slice across the batch, the smallest data shard that can be passed to an accelerator is the $\Theta(n^2)$ sized image itself. Hence, we would need to distribute $\Theta(bn^2)$ pieces of data across $A$ accelerators, where each data shard must be at least size $\Theta(n^2)$.
> > >
> > > Consequently, our approach in this particular example is strictly better than the direct approach that you mentioned.

---

### Official Review · Reviewer_9skV · 2024-11-08

**Soundness:** 2
**Presentation:** 1
**Contribution:** 2
**Rating:** 3
**Confidence:** 3

**Summary:**

This paper presents three efficient gradient clipping methods for convolution models, which aims to mitigate the computational bottleneck of per-sample gradient norm estimation in differentially private stochastic gradient descent. Two of the proposed methods use in-place operations, while the third uses the connection between Fourier transforms and convolution layers. The experimental results show that the proposed methods outperform other existing methods both in running time and memory consumption.

**Strengths:**

1. This paper develops three improved efficient methods of the gradient clipping step specifically designed for convolutional layers, addressing the computational bottleneck in differentially private stochastic gradient descent (DP-SGD).
2. The analysis for the third method is novel, which utilizes properties of circulant matrices to derive an algorithm that scales effectively with both the number of model parameters and batch size, demonstrating superior performance in high-dimensional settings.

**Weaknesses:**

1. The contributions of the proposed methods are not clear, especially the first two methods which seem only incorporating the in-place operations on the current exsting works (FastGradClip and Ghost Clipping), without significant conceptual advancements.
2. The claimed improvements in computation time are not fully substantiated. In Setting #1 of the section 4 (numerical results), the authors note that “our in-place ghost-norm algorithm was performing significantly worse than the rest of the methods and was increasing the computation time of the experiment significantly, so we have not displayed it”. This raises concerns about whether the proposed methods genuinely enhance computational efficiency.
3. In the Figure 4.1, Figure 4.2, and Figure 4.3 (running time analysis), the y-axis appears negative values, unclear why the running time metric would be negative. Also, the negative values appear in the memory consumption metric in Figure 4.2.
4. The experimental setup lacks clarity, particularly regarding the structure and the number of layers in the CNNmodel used for evaluation. This makes it difficult to assess the generalizability of the results.
5. The paper does not adhere to the ICLR formatting guidelines, which affects readability and may impact the overall presentation quality.

**Questions:**

1. Could you clarify how the first two methods differ fundamentally from existing techniques like FastGradClip and Ghost Clipping beyond using in-place operations?
2. In the numerical results section, why was the in-place ghost-norm algorithm excluded from some plots due to poor performance? Does this indicate a significant limitation of the method?
3. Could you explain why the y-axis in Figures 4.1, 4.2, and 4.3 shows negative values? Is this an error, or is there a specific reason for representing the data this way?
4. Could you provide more details on the CNN architecture used in your experiments (e.g., number of layers, types of layers)? How might the results change with different CNN architectures?
5. Can you provide more insights on the scalability of Method 3 when applied to real-world large-scale vision datasets, especially regarding runtime and memory overhead?

---

> ### Author Response · Authors · 2024-11-21
>
> We would like to thank the reviewer for their review. Please find our response to the weaknesses and the questions you have mentioned below.
>
> Weakness 1: While the first two methods use existing formulas in the literature to derive gradient norm estimates, our focus for these methods was on more efficient implementations in practice. More specifically, prior work involved “unrolling” the gradients to large matrices and performing matrix multiplication, which required storing large auxiliary matrices in order to perform the computations in these operations. On the other hand, our approach is to analyze the updates as abstract linear operators and, consequently, develop a more efficient scheme for evaluating them.
> Note that one of the relevant topics listed on the ICLR website is “implementation issues” for which this paper is relevant (see the third last bullet point on https://iclr.cc/Conferences/2025).
>
>
> Weakness 2: Since we ran relatively small-scale experiments, we believed that it was not informative to include the performance of the in-place ghost norm algorithm, at the cost of extra consumption of resources. The main advantage of these two methods is the storage efficiency compared to prior work. We will include this runtime in the next version of our work.
>
> Weakness 3: The y-axes of our plots have negative values since we are plotting the log of the runtime and the memory consumption. This allows us to view improvements on a multiplicative scale rather than an additive one.
>
>
>
> Weakness 4: In the experimental setting, we only test a single layer of the CNN since the complexity of our approach (and of prior work) scales linearly with the number of layers. Moreover, there are no computations involving more than one layer at a time. We believe that this comparison gives a more accurate understanding of the running time and memory storage improvements of our methods. The parameters of the particular layer we use for the experiment are provided in Section 4 (please refer to $d_{in}, d_{out}, n_{in}, n_{out}, d_k$.)
>
> Weakness 5: We apologize for using the wrong template. We have now uploaded a new version of our manuscript using the correct template.
>
> Question 1: Please see our answer to weakness 1.
>
> Question 2: Please see our answer to weakness 2.
>
> Question 3: Please see our answer to weakness 3.
>
> Question 4: Please see our answer to weakness 4.
>
> Question 5: We are unsure about what you mean by scalability in this context. In terms of memory and runtime scaling, we have asymptotic bounds in Table 1.1 (page 2) and further analysis in section 3.3.

---

> ### Comment · Reviewer_9skV · 2024-11-27
>
> Thank you for providing the author response. I would keep my scores.

---

### Official Review · Reviewer_YYgv · 2024-11-08

**Soundness:** 3
**Presentation:** 2
**Contribution:** 2
**Rating:** 3
**Confidence:** 2

**Summary:**

The paper proposes three new methods for efficiently gradient-clipping in differentially private stochastic gradient descent (DP-SGD) for convolutional neural network models. The key contributions are:

1. They propose two in-place algorithms (Algorithms 3.1 and 3.2) that compute the gradient norm using only O(1) memory storage, in contrast to the linear memory usage depending on model dimensions of prior methods. They also propose a Fourier-based algorithm (Algorithm 3.3) that leverages the structure of convolution layers to compute the gradient norm more efficiently, with a runtime that scales better with the input/output dimensions.
2. They provide theoretical analysis demonstrating the runtime and memory advantages of the proposed methods compared to prior work, depending on the specific model architecture.
3. The experiments shows the advantage of the proposed algorithms across different regimes of model size and channel counts.

**Strengths:**

1. The proposed algorithms provide significant improvements in memory usage compared to prior art, with theoretical and empirical support.
2. The analysis is rigorous, leveraging properties of circulant matrices and Fourier transforms in a novel way for the gradient clipping problem.
3. The empirical evaluation covers a range of relevant settings and provides clear comparisons to prior work.
4. The methods are general and can be applied to a variety of convolutional neural network architectures.

**Weaknesses:**

1. The evaluation is limited to just the gradient clipping step, and does not include full DP-SGD training experiments. I understand that the only modified part is gradient clipping, but demonstrating the end-to-end impact would be a more straightforward message. Will the proposed method lead to fast overall convergence, and how is the generalization performance?
2. While the paper provides three algorithms, it does not give any insight on how one should choose between the different proposed methods, given a specific model architecture and resource constraints.

**Questions:**

Please see weaknesses.

---

> ### Author Response · Authors · 2024-11-21
>
> We would like to thank the reviewer for their review. Please find our response to the weaknesses and the questions you have mentioned below.
>
> Indeed, since the only step we modify is the gradient clipping step, comparing layer-level performance is more direct than end-to-end performance. Moreover, if we run DP-SGD for the same number of steps and under different clipping subroutines, then the utility of the algorithm will be unaffected by the change since the clipped gradients will be identical. Nevertheless, we will include experiments with end-to-end training in the next version of our work.
>
>
> We respectfully disagree with the reviewer’s comment. In Section 3.3 we provide a comparison of the runtime of the different algorithms in terms of the architecture of the underlying layer of the CNN. Moreover, in Section 4 we illustrate different regimes in which different algorithms perform better. If there are any further comparisons the reviewer would like to see, we would be happy to provide them.

---

> ### Comment · Reviewer_YYgv · 2024-11-25
>
> Thank you for your rebuttal. It solves my second concern, but for first concern, I think more comprehensive experiments, e.g., generalization performance are needed.

---

> > ### Author Response · Authors · 2024-11-25
> >
> > Note that the iterates generated by our method and (unmodified) DP-SGD are the same because the clipped gradients between the two methods are the same.
> >
> > Consequently, the convergence, generalization (to unseen test data), and utility are identical between our algorithm and (unmodified) DP-SGD.

---

### Official Review · Reviewer_Gift · 2024-11-09

**Soundness:** 3
**Presentation:** 2
**Contribution:** 2
**Rating:** 5
**Confidence:** 3

**Summary:**

The authors proposed several algorithms to reduce the space and time complexity of calculating the (squared) norm of gradients particularly for convolutional layers in DP-SGD. They provided theoretical analyses of each algorithm's complexity and verified their effectiveness across various settings.

**Strengths:**

- Rigorous theoretical analyses on the space and time complexity of each algorithm
- Improved performance across different settings

**Weaknesses:**

- The authors should clearly specify the problem they are addressing and what lies outside their scope, allowing readers to understand and appreciate the contribution of this work from a broader perspective. Specifically:

    - The authors should outline the general framework for per-sample gradient clipping. For example, in the OPACUS framework, there is a "for" loop that iterates over different layers: (i) the per-sample gradient accumulates across layers; and (ii) different types of layers (e.g., convolutional layers, fully connected layers) use distinct implementations to calculate the gradient norm. In this sense, the authors are tackling a low-level problem and do not address issues such as parallelizing different samples.

    - The authors should clarify the nuanced difference between “time complexity” and “actual runtime”. Practically, the computational complexity of per-sample gradient clipping is only a constant factor less efficient compared to standard training, rather than being linearly dependent on batch size.

- The proposed algorithm is valuable only if it outperforms current best practices. Unfortunately, the authors only use Bu et al. (2022) as the baseline and do not compare against Bu et al. (2023b), which they acknowledge as state-of-the-art, nor do they consider widely used frameworks like OPACUS and JAX_privacy. Notably, OPACUS provides a tailored solution for per-sample gradient clipping in convolutional modules. This comparison seems essential, and I strongly encourage the authors to discuss whether their implementation of the algorithms could be integrated into these existing frameworks.

**Questions:**

- The term "additional storage" is unclear. My understanding is that storing gradients requires $O(d)$ space, where $d$ represents the number of parameters in the layer. Could you clarify if "additional storage" refers to something beyond this baseline requirement, and that it appears in all prior work?

- I'm unfamiliar with "ghost clipping", and I assume many readers will be as well. It would be beneficial to provide a clear explanation of this concept in the paper.

---

> ### Author Response · Authors · 2024-11-21
>
> We would like to thank the reviewer for their comments. Below we address the points they raised.
>
> In the next version of our manuscript, we will give a description of DP-SGD and describe where our improvements fit inside the DP-SGD. For a preview, we give an outline of this description below.
> "DP-SGD is a variant of SGD in which the per-example gradients in the SGD update are replaced by a set of privatized per-example gradients. These privatized gradients are obtained by (i) projecting each of the original per-example gradients onto a Euclidean ball of some radius C (commonly known as the L2 clip norm) and (ii) adding Gaussian noise Z to the sum of the projected gradients with variance depending on C and the size of the input batch used to compute the original gradients.
> One efficient method of projecting the per-example gradients is to compute the per-example gradient norms and compute the gradient of a weighted loss function whose weights depend on the norms and radius C. For training machine learning models specifically, the computation of the squared gradient norms can be decomposed by the different layers in the model. Because of this decomposition, there have been several works that proposed efficient methods for computing gradient norms of specific model layers.
> This work develops several methods for efficiently computing the gradient norms of the convolutional layer under different parameter regimes. These methods can be directly incorporated in optimization libraries that implement per-layer gradient and gradient norms operations, such as Opacus, tensorflow-privacy, and jax-privacy, and benefit from these libraries resources, such as parallelization."
>
> Regarding the nuances between “actual runtime” and “time complexity”, we disagree about the (constant-factor) equivalence between standard training and per-example clipping. For this discussion, let us consider the non-private/private gradient computation step, and suppose the model under consideration has N model weights and that input batches are of size B.
>
> In standard (non-private) training, one typically computes the gradient of a scalar loss function with respect to model weights, incurring roughly N storage/runtime cost (auxiliary quantities during this computation are typically on the same order of magnitude).
>
> On the other hand, in the (naive) implementation of DP-SGD, the training step consists of a for-loop over the B per-example (or per-sample) losses where the gradients of each per-example losses are materialized, clipped (by projection), and then aggregated. Since materialization of the gradients and computing the subsequent norms is an Ω(N) operation, the storage/runtime complexity is Ω(N*B) operation, which scales linearly with the batch size.
>
> The algorithm in Bu et al. (2023b) is a hybrid of the naive per-example gradient clipping algorithm and the ghost norm clipping algorithm for convolution layers. Specifically, the Bu et al. algorithm conditionally invokes either the naive or ghost norm algorithm for each convolution layer depending on the number of channels, input dimension, output dimension, and kernel dimension of that layer (see Section 3 of Bu et al. (2023b)). Consequently, we believe a fair comparison would be to compare each of the clipping algorithms in Bu et al. (2023b) against our algorithms in the regimes that they perform the best in (see the three settings in our Section 4). This is equivalent to making the comparison to the clipping algorithms in Bu et al. (2022) over the best performing regimes.
>
> Opacus code for convolution layers (see https://github.com/pytorch/opacus/blob/main/opacus/grad_sample/conv.py) applies the ghost norm clipping algorithm for linear/dense layers under the framework that the convolution layer is a linear layer whose inputs are the “unfolded” versions of the input images. Thus, the performance of the Opacus implementation would be similar to that of what we label “Bu et al. (2022) ghost norm” in Figures 4.1-4.3 of the current version of our paper. In the next version, we will add a detailed comparison of our approach to existing implementations in Opacus and JAX privacy.
>
> In terms of integration efforts, we do not see any issues that prevent us from integrating with existing fast clipping software packages such as Opacus and TensorFlow privacy. For convenience, we give some details of what the integration would look like with respect to the interfaces of the aforementioned packages.
>
> In Opacus (https://github.com/pytorch/opacus/blob/main/opacus/grad_sample/conv.py), the interface of the per-example (convolution) gradient norm computer consumes (i) metadata about the layer, (ii) layer inputs (activations), and (iii) backpropagation gradients. Items (i)–(iii) are clearly sufficient for implementing our Algorithms 3.1–3.2 whereas the implementation of Algorithm 3.3 only requires an additional Fast Fourier Transform oracle, for which we can use NumPy’s implementation (numpy.fft).

---

> ### Author Response · Authors · 2024-11-21
>
> A similar interface is used for TensorFlow privacy (https://github.com/tensorflow/privacy/blob/master/tensorflow_privacy/privacy/fast_gradient_clipping/registry_functions/dense.py), so we do not expect any issues.
>
> As far as we know, JAX Privacy does not have a framework for fast clipping. However, it does support the (naive) per-example clipping algorithm and, hence, we can integrate our Algorithm 3.1.
>
>
> The term of additional storage refers to the extra storage space needed to compute the norm of the gradients on top of the storage used by non-private SGD. More precisely, after the back-propagation step, non-private SGD has access to $x$ and $g$ in Algorithm 3.1 in our manuscript. We will elaborate on this in the next version of our manuscript.
>
>
>  “Ghost clipping” refers to a gradient norm computation trick introduced by Goodfellow in 2015 for efficiently computing the gradient norms of fully connected layer weights. Our work uses a similar trick in Algorithm 3.1 that avoids directly computing the norm. In the current version of our work we give a reference to Goodfellow’s paper, but we will elaborate more in the next version of our draft.

---

> ### Comment · Reviewer_Gift · 2024-11-21
>
> Thanks for the response.
>
> - I generally like the proposed outline; it will help clarify how your work fits within the broad DP-SGD framework. A few additional comments: 1) I didn’t quite follow the part about "computing the gradient of a weighted loss function whose weights depend on the norms and radius C" — I thought computing the per-sample gradient norm would suffice. 2) Including a flow chart alongside the outline would be great. 3) It would be even better to highlight the challenges not addressed by your work (e.g., parallelization of samples, as this is mentioned in the introduction).
>
> - I think you’re still referring to "time complexity", as the discussion pertains to a naive, sequential implementation of per-sample gradient clipping. In practice, however, parallelization is applied at this step. This is precisely why I think it’s important to distinguish between "worst-case time complexity" and "actual runtime".
>
> - It would be helpful to discuss the choices of baseline selection and include comparisons with Opacus in the revised version. I believe this will further strengthen the paper.
>
> - Thanks for answering my remaining questions.

---

> > ### Author Response · Authors · 2024-11-22
> >
> > Thank you for the response. Please find some additional clarifications below:
> >
> > > 1) I didn’t quite follow the part about "computing the gradient of a weighted loss function whose weights depend on the norms and radius C" — I thought computing the per-sample gradient norm would suffice.
> >
> > In the Ghost Clipping framework, once the gradient norms are computed, we must find a way to aggregate the clipped gradients **without** materializing the actual gradients. This can be done by creating a special loss function whose gradient is equal to aggregation of the clipped gradients. It turns out that this special loss function can be obtained by creating a weighted loss function from the original per-example loss functions, whose weights depend on C and the gradient norm. In this way, we are replacing the computation of materializing the per-example gradients with a back-propagation step.
> >
> > For more precise details, you can see Section 4 of the following paper:
> >
> > * Kong, W., & Munoz Medina, A. (2024). A unified fast gradient clipping framework for DP-SGD. *Advances in Neural Information Processing Systems*, 36.
> >
> > > 2) Including a flow chart alongside the outline would be great. 3) It would be even better to highlight the challenges not addressed by your work (e.g., parallelization of samples, as this is mentioned in the introduction).
> >
> > We will add these to the revision.
> >
> > > I think you’re still referring to "time complexity", as the discussion pertains to a naive, sequential implementation of per-sample gradient clipping. In practice, however, parallelization is applied at this step. This is precisely why I think it’s important to distinguish between "worst-case time complexity" and "actual runtime".
> >
> > Indeed, different implementations will lead to different (actual) runtimes (due to parallelization or other optimizations).
> >
> > However, in practice, we believe that large LLMs (on the order of billions of parameters) cannot actually perform parallelization effectively unless they are trained on a much larger number of accelerators compared to non-private training. More specifically, if a user is restricted to only a single accelerator and the LLM parameters can only fit on a single accelerator, then parallelization of the per-sample gradients would not be feasible. We will add additional information in the revision to clarify this point (and the other extreme where we have enough accelerators to perform fully parallelized per-example gradient clipping).
> >
> > > It would be helpful to discuss the choices of baseline selection and include comparisons with Opacus in the revised version. I believe this will further strengthen the paper.
> >
> > We will add a discussion on how the baselines were selected and try to set up some experiments with Opacus.

---

### Meta-Review · Area_Chair_Akhq · 2024-12-26

**Metareview:**

The paper proposes several algorithms aimed at reducing the space and time complexity involved in calculating the (squared) norm of gradients, specifically for convolutional layers in Differential Privacy Stochastic Gradient Descent (DP-SGD). The authors provide theoretical analyses of the complexity of each algorithm and validate their effectiveness through empirical experiments in various settings, demonstrating improvements in computational efficiency.

Based on the reviews and discussions, there was a consensus that the paper is limited in novelty, as the proposed methods largely rehash existing techniques with minimal conceptual advancements. Claims of computational and memory efficiency are weakly substantiated, with significant issues in the experimental setup and presentation, including unclear results, errors in figures, and insufficient detail on the evaluation methodology. These shortcomings undermine the paper's contribution and impact.

**Additional Comments On Reviewer Discussion:**

NA

---

### Decision · Program_Chairs · 2025-01-22

Reject